# *TERT* expression is associated with metastasis from thin primaries, exhausted CD4+ T cells in melanoma and with DNA repair across cancer entities

Christina Katharina Kuhn[1]*, Jaroslawna Meister[1,2], Sophia Kreft[3], Mathias Stiller[4], Sven-Holger Puppel[1], Anne Zaremba[3], Björn Scheffler[5], Vivien Ullrich[5], Torsten Schöneberg[1,6], Dirk Schadendorf[3], Susanne Horn[1,3]*

1 Rudolf Schönheimer Institute of Biochemistry, University of Leipzig, Medical Faculty, Leipzig, Germany, 2 Institute for Clinical Diabetology, German Diabetes Centre, Leibniz Centre for Diabetes Research at Heinrich Heine University Düsseldorf, Düsseldorf, Germany, 3 Department of Dermatology, University Hospital Essen, University Duisburg-Essen, and German Cancer Consortium Partner Site Essen/Düsseldorf, Essen, Germany, 4 Institute of Pathology, University of Leipzig Medical Center, Leipzig, Germany, 5 DKFZ-Division Translational Neurooncology at the West German Cancer Center, University Hospital Essen/ University of Duisburg-Essen, Essen, Germany, 6 School of Medicine, University of Global Health Equity, Kigali, Rwanda

* Susanne.Horn@medizin.uni-leipzig.de (SH); Christina.Kuhn@medizin.uni-leipzig.de (CKK)

**Data Availability Statement:** The data used in this study are available within this paper and its

## Abstract

Telomerase reverse transcriptase (*TERT*) promoter mutations occur frequently in cancer, have been associated with increased *TERT* expression and cell proliferation, and could potentially influence therapeutic regimens for melanoma. As the role of *TERT* expression in malignant melanoma and the non-canonical functions of TERT remain understudied, we aimed to extend the current knowledge on the impact of *TERT* promoter mutations and expression alterations in tumor progression by analyzing several highly annotated melanoma cohorts. Using multivariate models, we found no consistent association for *TERT* promoter mutations or *TERT* expression with the survival rate in melanoma cohorts under immune checkpoint inhibition. However, the presence of CD4+ T cells increased with *TERT* expression and correlated with the expression of exhaustion markers. While the frequency of promoter mutations did not change with Breslow thickness, *TERT* expression was increased in metastases arising from thinner primaries. As single-cell RNA-sequencing (RNA-seq) showed that *TERT* expression was associated with genes involved in cell migration and dynamics of the extracellular matrix, this suggests a role of *TERT* during invasion and metastasis. Co-regulated genes found in several bulk tumors and single-cell RNA-seq cohorts also indicated non-canonical functions of *TERT* related to mitochondrial DNA stability and nuclear DNA repair. This pattern was also evident in glioblastoma and across other entities. Hence, our study adds to the role of *TERT* expression in cancer metastasis and potentially also immune resistance.

supplementary files. Previously published, processed NGS data can be accessed at https://tools.hornlab.org/cru337phenotime/.

**Funding:** This work was funded by the Deutsche Forschungsgemeinschaft (DFG, German Research Foundation, HO 6389/2-2, SCHA 422/17-2, 'KFO 337' - 405344257). The funders had no role in study design, data collection and analysis, decision to publish, or preparation of the manuscript.

**Competing interests:** The authors have declared that no competing interests exist.

**Abbreviations:** anti-CTLA4, Antibodies for cytotoxic T-lymphocyte-associated protein 4; BRAF, V-raf murine sarcoma viral oncogene homolog B1; CCLE, Cancer cell line encyclopedia; CD4+/CD8+, Cluster of differentiation 4/8; Cox PH, Cox proportional-hazard ratio; DNA2, DNA replication helicase/nuclease 2; ETS, Erythroblast transformation specific (transcription factor); FPKM, Fragments per kilobase of transcript per million; GAPB, GA-binding protein; GBM, Glioblastoma multiforme; GTEx, Genotype-tissue expression project; HAVCR2/TIM3, Hepatitis A virus cellular receptor 2/T-cell immunoglobulin and mucin-domain containing-3; HRAS, Harvey rat sarcoma viral oncogene; ICI, Immune checkpoint inhibition; KRAS, Kirsten rat sarcoma viral oncogene; MAPK, Mitogen-activated protein kinase; MEK, Mitogen-activated kinase; Myc, Myelocytomatosis, MYC proto-oncogene, bHLH transcription factors; NF-kB, Nuclear factor kappa B (transcriptions factor); NF1, Neurofibromatose type 1; NRAS, Neuroblastoma ras viral oncogene homolog; OS, Overall survival; PCAWG, Pan-cancer analysis of whole genomes cohort; PCDC1/PD1, Programmed cell death protein 1; PFS, Progression free survival; ROS, Reactive oxygen species; scRNA-seq, Single-cell RNA-sequencing; SKCM, Skin cutaneous melanoma cohort; TCF, Ternary complex factor; TERT, Telomerase reverse transcriptase; TGF-beta, Transforming growth factor beta; TIGIT, T cell immunoreceptor with Ig and ITIM domains; TMB, Tumor mutational burden; TPM, Transcripts per million; Wnt/β-catenin, "Wingless-related integration site" Wnt/beta-catenin pathway.

## Introduction

Telomere maintenance mechanisms play a critical role in cellular survival as they are known to enhance the proliferative capacity of cancer cells [1]. Frequent mutations in the promoter region of the telomerase reverse transcriptase (*TERT)* gene were shown to be associated with the upregulation of telomerase expression [2–4]. *TERT* promoter mutations have been proposed as independent prognostic markers in non-acral cutaneous melanoma, since they correlated in a multivariate analysis with decreased overall survival (OS) without analysis of therapy [5]. The co-occurrence of *BRAF* or *NRAS* mutations with *TERT* promoter mutations was also prognostic for poor disease-free survival in multivariate analyses in primary melanoma [6]. *TERT* promoter mutations may serve as predictive biomarkers as they were independently associated with improved survival in patients receiving BRAF/MEK inhibitor therapy in univariate and multivariate analyses [7, 8]. Also, patients undergoing anti-CTLA4 therapy showed better overall survival with *TERT* promoter mutations, implying its potential as a predictive marker for immune checkpoint inhibitor (ICI) therapy [9]. Also, *TERT* expression has been suggested to be an independent prognostic marker associated with poor survival in human cancer patients beyond melanoma [10–12]. However, many variables, such as mutational load, tumor heterogeneity, and prior therapies, could be appended to existing multivariate models to test additional potentially stronger predictors.

The extent to which genetic alterations of *TERT* play a role in cancer progression and the molecular mechanisms affected by reactivation of *TERT* transcription vary widely among cancer entities, making it crucial to understand the underlying mechanisms [13]. One way cancer cells regain their proliferative ability is by forming a new ETS binding motif at the promoter mutation [2]. This binding motif attracts transcription factors, such as TCF1, Myc, Wnt/β-catenin, NF-kB, and GAPB, which are responsible for regulating numerous cellular processes including tumorigenesis [2, 14]. The canonical function of *TERT* is telomere maintenance, however, non-canonical functions of *TERT* have been discussed, e.g. related to reactive oxygen species (ROS), mitochondria and aging. TERT is imported into the mitochondria, where it binds mitochondrial DNA, protecting it from oxidative stress–induced damage, reducing apoptosis in human endothelial cells [15]. Furthermore, besides the full-length expressed *TERT*, alternatively spliced isoforms exist, which lack telomerase activity and may account for up to 10% of endogenous *TERT*. These are continuously expressed after telomerase silencing during human embryonic development [16] and are potentially associated with telomere-independent effects [17].

Here, we test if *TERT* promoter mutations and *TERT* expression are associated with OS and progression free survival (PFS) under ICI, as well as clinicopathological and demographic parameters in multiple highly annotated melanoma cohorts with exome- and transcriptome sequencing. To examine the non-canonical functions of *TERT*, we investigate associations with ROS-related genes, hypoxia, chromothripsis, and *TERT* isoform expression. Using molecular pathway analyses, we replicate known and identify new signaling pathways regulated with genetically modified *TERT* in bulk tumors, cell lines, and single-cells. Tumor immune signatures are estimated to elucidate the role of *TERT* during tumor rejection. Finally, our analyses are extended to additional cancer entities from the Pan-Cancer Analysis of Whole Genomes (PCAWG) cohort and glioblastoma (GBM) to determine the signaling pathways co-regulated with *TERT* across entities.

## Materials and methods

### Next generation sequencing data analysis

We analyzed several bulk RNA-seq cohorts with respective survival data (Table 1, Transcriptomics in S1 File). These included five melanoma cohorts with samples taken prior to or early-

**Table 1. Analyzed cohorts.**

| Name | Datatype | Expression Unit | Source | *TERT* Promoter Status known | Cancer Entities | Immune Checkpoint Blockade Therapy | PMID | Mito data |
|---|---|---|---|---|---|---|---|---|
| SKCM | Bulk RNA-Seq | TPM | Tissue | Subset | Melanoma | No | 26091043 | yes |
| Gide | Bulk RNA-Seq | TPM | Tissue | No | Melanoma | Yes | 30753825 | no |
| Riaz | Bulk RNA-Seq | TPM | Tissue | No | Melanoma | Yes | 29033130 | no |
| Van Allen | Bulk RNA-Seq | TPM | Tissue | Yes | Melanoma | Yes | 26359337 | no |
| Liu | Bulk RNA-Seq | TPM | Tissue | No | Melanoma | Yes | 31792460 | no |
| Hugo | Bulk RNA-Seq | TPM | Tissue | No | Melanoma | Yes | 26997480 | no |
| Li | Panel Seq | not applicable, mutations | Tissue | Yes | Multiple | Yes | 32810393 | no |
| Jerby-Arnon | Single-Cell RNA-Seq | TPM | Tissue | No | Melanoma | Yes | 30388455 | no |
| PCAWG | Bulk RNA-Seq | uqv2 FPKM for genes, TPM for isoforms | Tissue | Subset | Multiple | No | 32025007 | yes |
| CCLE | Bulk RNA-Seq | TPM | Cell Culture | Subset | Multiple | No | 31068700 | no |
| Klughammer | Bulk RNA-Seq | RPKM | Tissue | No | Glioblastoma | No | 30150718 | no |

TPM = transcripts per million; mito data = mitochondrial data available; Uqv2 FPKM = upper quartile normalized fragments per kilobase of transcript per million mapped reads; RPKM = reads per kilobase of transcript per million mapped reads

on ICI therapy (Gide, Hugo, Liu, Riaz, Van Allen cohorts), one melanoma cohort without ICI therapy (Skin Cutaneous Melanoma cohort, SKCM), two cross-entity datasets: cell lines from the Cancer Cell Line Encyclopedia (CCLE), and bulk tumors of the Pan-Cancer Analysis of Whole Genomes (PCAWG). We further included melanoma single-cell RNA-seq (scRNA-seq) and glioblastoma tumors in our workflow (S1 Figin S1 File). In brief, we extended the available gene expression levels with additional metadata, as described below and in the S1 File. We annotated published *TERT* promoter mutations in the RNA-seq datasets of two melanoma cohorts (Van Allen, SKCM) and two cross-entity datasets (CCLE and PCAWG, *TERT* promoter mutation calls in S1 Material). We added immune signatures for the six melanoma bulk RNA-seq cohorts using TIMER 2.0 [18] (Immune signatures in S1 Material). Isoform analysis was performed for the Van Allen, SKCM and PCAWG cohorts. To analyze mitochondria-related features in the PCAWG and SKCM cohorts, we further integrated measurements for hypoxia, chromothripsis [19] and mitochondrial mutations [19, 20]. Hypoxia scores for PCAWG were determined as in Buffa et al. 2010 [21], and hypoxia scores for the SKCM cohort were obtained from the cBioPortal (www.cbioportal.org/study/clinicalData?id=skcm_tcga_pan_can_atlas_2018). We reanalyzed the MSK IMPACT Clinical Sequencing Cohort (dataset "MSKCC, Nat Med 2017") [9] as obtained from www.cbioportal.org annotating the corresponding therapy data from Samstein [22].

## Enrichment analysis

Enrichment analysis of top genes associated with *TERT* promoter mutation and *TERT* expression using bulk RNA-seq cohorts was conducted using the gene set enrichment analysis tool https://maayanlab.cloud/Enrichr/ [23] with default settings. For enrichment analysis in the melanoma single-cell context, the top 100 genes associated with *TERT* expression in the scRNA-seq dataset were queried with Enrichr using default settings. *TERT* promoter mutation status was not available for single-cell data. The top ten results of each enrichment set from gene ontology pathways, ontologies and cell types were then used for discussion of the most frequently occurring terms.

### *TERT* isoform analysis

Short names, coding information, biotype, and transcript lengths of *TERT* isoforms were collected from Ensembl (GRCh38.p13) using ENSG gene identifiers (*TERT* isoform information in S1 Material). The presence of the mitochondrial targeting protein sequence 'MPRAPRCRAVRSLLRSHYRE' [24] in the translated isoforms was analyzed using the UCSC Human Genome Browser. In detail, we verified whether transcripts included the targeting sequence by aligning the protein sequence with the genome (GRCh38.p13) and comparing the start/stop positions with the isoform exons. The isoform expressed in normal control tissues was assessed using the GTEx portal (https://gtexportal.org/home/gene/TERT#gene-transcript-browser-block, S2 Fig in S1 File). We then summed over the measured transcripts per million (TPM) of all isoforms and calculated the proportion of isoforms with mitochondrial targeting sequence. Data on isoforms lacking the targeting sequence were available only for the CCLE and Van Allen cohorts.

### Statistics

Correlation of *TERT* mRNA expression, isoform expression and clinical variables was tested using Spearman correlations. For comparison between samples with or without *TERT* promoter mutation, two-sided Wilcoxon rank tests were applied. The correlates associated with *TERT* expression or *TERT* promoter mutations were ranked by p-value with false discovery rate control, and overlaps of the gene lists from several cohorts were performed. For survival analysis, the cohorts were divided into low vs. high *TERT* expression groups, with approximately one-third of the patients showing high *TERT* expression. If available, cohorts (SKCM and Van Allen cohort) were also divided into two groups by *TERT* promoter mutation absence/presence. Kaplan-Meier survival curves, log-rank tests, multivariate Cox Proportional-Hazards (Cox PH) analyses, and odds ratios were computed using R v.4.0.1 [25].

### Data availability statement

The data used in this study are available within this manuscript and its supplementary files. Previously published, processed next generation sequencing data can be accessed at https://tools.hornlab.org/cru337phenotime/.

## Results

In each of the following sections, we start at the genomic level by analyzing the role of *TERT* promoter mutations in malignant melanoma and extend the analysis to the transcriptomic level by examining the impact of *TERT* expression on clinical and molecular associations.

### Association of *TERT* promoter mutation status and *TERT* expression with survival

*TERT* promoter mutations and *TERT* expression are proposed as prognostic biomarkers in melanoma and *TERT* promoter mutations might be predictive under anti-CTLA4 therapy [5, 6, 9, 26, 27]. Therefore we investigated the impact of both *TERT* alterations using univariate and multivariate survival analyses based on six bulk RNA-seq cohorts of metastatic melanoma treated with ICI (Gide, Riaz, Van Allen, Liu, Hugo cohorts) and without targeted therapy or ICI (SKCM). Our analysis showed that *TERT* promoter mutations and *TERT* expression were not associated with survival (S3-S5 Figs, S1 Table). PFS under anti-PD1 immunotherapy with pembrolizumab appeared decreased with higher *TERT* expression in only one analyzed cohort with small sample size (n = 8 vs. 24, Gide cohort, PFS log rank p = 0.038), however, it was not

significantly associated in multivariate Cox PH including age, sex and *BRAF* mutation status (p = 0.071, S1A Table).

In contrast to our finding, an association of *TERT* promoter mutations with survival under anti-CTLA4 therapy was described previously [9], however, no multivariate model was attempted. Therefore, we reanalyzed the melanoma subset of the MSK Impact 2017 cohort [9] and the corresponding therapy data and included tumor mutational burden (TMB), sex, and *BRAFV*600 variants in a multivariate model. The previously described association of *TERT* promoter mutations with survival still holds, albeit with reduced significance (p = 0.015 compared to the previous univariate analysis p = 0.0007, S6A Fig in S1 File).

### *TERT* expression and increased intratumoral CD4+ T helper type 1 cells

To investigate the role of TERT during tumor rejection, we correlated immune signatures with TERT expression. The CD4+ T helper type 1 cell signature repeatedly increased with *TERT* in five of the six cohorts analyzed here (Fig 1A, S2 Table). However, within a patient population with high CD4+ T helper type 1 cell estimates, survival did not differ with elevated *TERT*

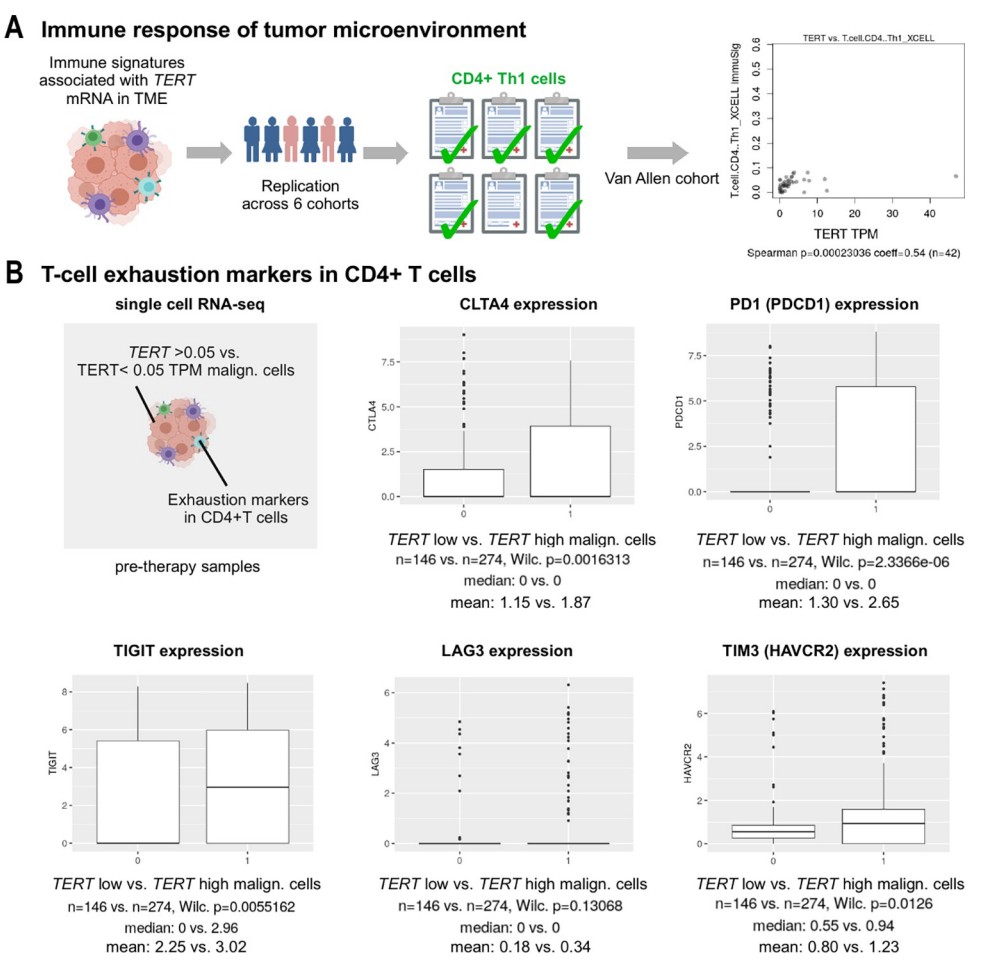

**Fig 1. *TERT* expression and immune signatures.** (A) Five out of six cohorts showed correlation of *TERT* expression and a CD4+ Th1 T cell immune signature (xCell assay). (B) Expression of T cell exhaustion markers in single CD4+ T cells of patients with low vs. high *TERT* expression in malignant tumor cells ($<$/$>$0.05 TPM). Jerby-Arnon cohort, pre-therapy samples. Th1: T-helper type 1; TME: tumor microenvironment; TPM: transcripts per million; xCell: xCell cell types enrichment analysis. The figure was created with BioRender.com.

expression (S7 Fig in S1 File). To find a possible interaction of CD4+ T cells, we tested the presence of potential exhaustion markers [28]. Indeed, *CTLA4, PDCD1I*PD1, *HAVCR2*/TIM3, and *TIGIT* were expressed at higher levels (Fig 1B) in CD4+ T cells of tumors with high *TERT* expression of malignant cells.

## Primary tumor thickness

Our examination of clinical variables in association with *TERT* promoter mutations and expression was based on the two largest melanoma cohorts, the SKCM and Liu cohorts (S3A–S3C Table). There was no obvious difference in the primary tumor thickness between metastatic samples with vs. without *TERT* promoter mutations (Wilcoxon p = 0.508, Fig 2A). Notably, *TERT* expression was higher in metastases of tumors arising from thinner primaries (Spearman p = 0.01, coeff = -0.16, Fig 2A; Wilcoxon p = 0.011, Breslow >2 mm: 8.95 TPM vs. Breslow <2 mm: 11.89 TPM, Fig 2B).

## *TERT* in melanoma subtypes

As the mitogen-activated protein kinase (BRAF-MAPK) signaling pathway is frequently activated in melanoma [29], we analyzed *TERT* in melanoma subtypes with hotspot mutations in *BRAF, RAS (NRAS, KRAS, HRAS)* and *NF1* genes in the SKCM cohort. Melanomas with hotspot mutations carried *TERT* promoter mutations more often than triple wild-type melanomas (odds ratios >20, S4A Table). With respect to *TERT* expression, we found significantly lower *TERT* expression in *BRAF* hotspot mutants than in *RAS* hotspot mutants (Wilcoxon p = 0.0004, S4B Table). The group of patients with *TERT* promoter mutations showed down-

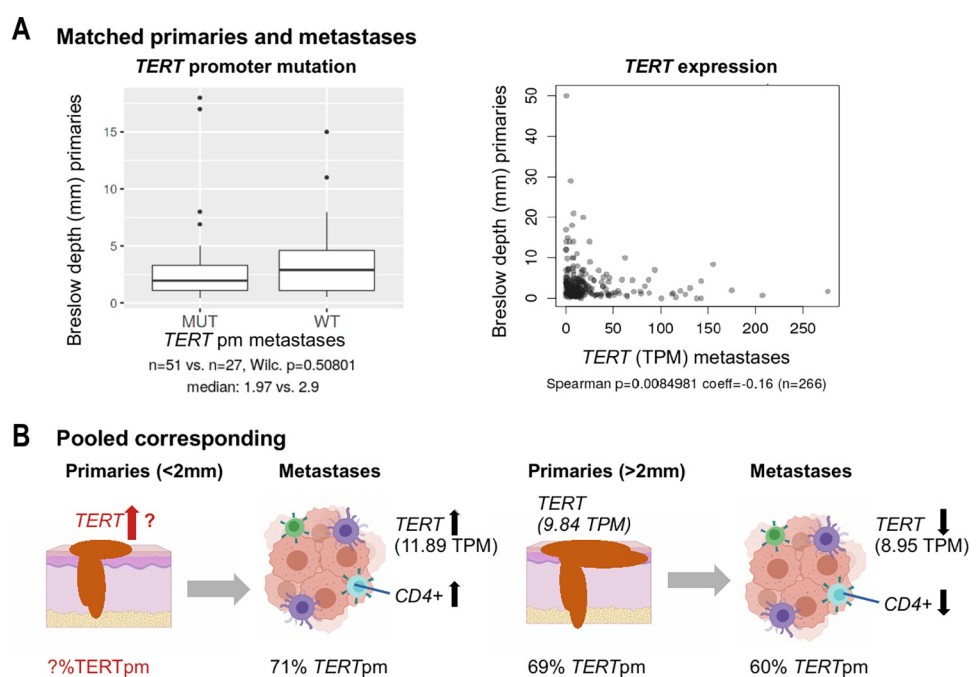

**Fig 2. Primary tumor thickness and *TERT* in corresponding metastases.** (A) Breslow depth of primaries with *TERT* promoter mutations and *TERT* expression measured in corresponding metastatic tumors of the SKCM cohort (without one outlier; *TERT* expression >600, for data including outliers see S8 Fig in S1 File; also see S3D Table). (B) *TERT* expression in metastatic tumors arising from thin (Breslow depth <2 mm) and thick tumors (>2 mm). Information on *TERT* alterations in thin primaries are not available. *TERT*pm: *TERT* promoter mutations; TPM: transcripts per million. The figure was created with BioRender.com.

regulated *TERT* expression in *BRAF* mutants and up-regulated *TERT* expression in *RAS* mutants (mutated *RAS*: Wilcoxon p = 0.024, mutated *BRAF*: Wilcoxon p = 0.004, S4C Table).

## Mutational load

A common pattern of *TERT* promoter mutations and *TERT* expression was the association with more nuclear mutational load. In the entire SKCM cohort, samples with *TERT* promoter mutations showed higher overall mutation counts (Wilcoxon p = 0.014, Fig 3A, not significant for the metastatic subgroup) and higher *TERT* expression was also associated with mutation counts (Spearman p = 0.002, coeff = 0.16, S3B Fig in S1 File, metastatic subgroup: Spearman p = 0.017, coeff = 0.14, Fig 3A), however, not in the Liu cohort (Spearman: 0.194, coeff = 0.12, S3A and S3B Table).

Moreover, in the SKCM cohort we did not find *TERT* promoter mutations associated with Breslow depth or age at diagnosis, and we did not find *TERT* expression associated with sex

**A** **Nuclear mutational load**

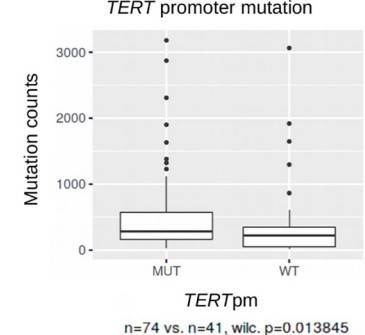
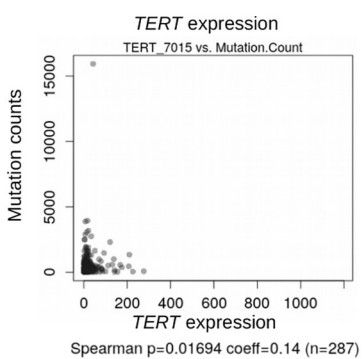

**B** **Pathway analysis**

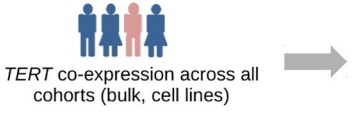
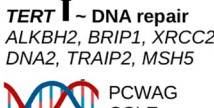
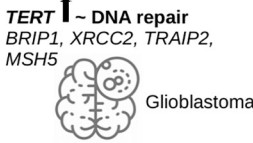

**C** **Mitochondrial mutational load and hypoxia**

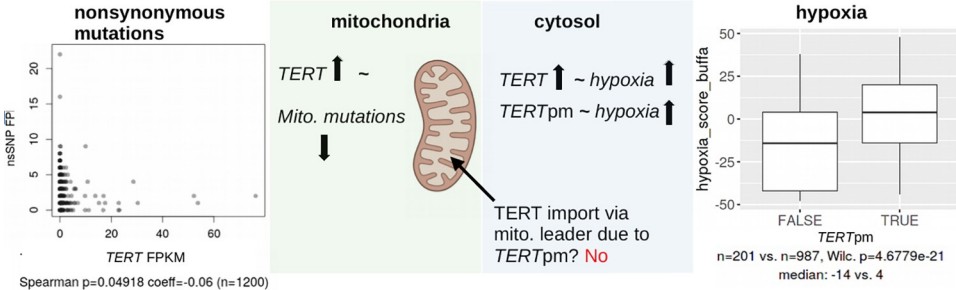

**Fig 3. Genetic associates of *TERT* alterations.** (A) *TERT* alterations and exome mutation load in the SKCM cohort. (B) *TERT* expression and DNA repair genes associated across all cohorts. (C) *TERT* alterations, mitochondrial mutation load and hypoxia (PCAWG cohort). TPM: transcripts per million; FPKM: fragments per kilobase of transcript per million fragments mapped; nsSNP: non-synonymous mitochondrial SNP; *TERT*pm: *TERT* promoter mutations. The figure was created with BioRender.com.

(SKCM, Liu), age at diagnosis (SKCM), or various clinicopathological variables of the Liu cohort, such as heterogeneity or total neoantigens (S3 Table).

## Gene expression associated with *TERT* alterations

At first, we replicated the previously described association of higher *TERT* expression levels in the presence of *TERT* promoter mutations in three independent cohorts (S5A Table). Since alternative splicing may result from mutation of the *TERT* promoter [17], we analyzed the expression of *TERT* isoforms. However, none of the isoforms were consistently regulated with *TERT* promoter mutation (S5A Table, S9 Fig in S1 File).

Next, we aimed to identify other cellular signaling pathways that may be driven by or co-regulated with either *TERT* promoter mutations (S6 Table) or *TERT* expression (S7 Table) followed by an enrichment analysis of associated gene expression. Associations with the presence of *TERT* promoter mutations revealed an overlap of 30 genes in the three bulk RNA-seq cohorts (in melanoma and across entities), of which six genes were consistently up- or down-regulated (up: *ARL17A*, *FBF1*, *KAZALD1*, *PRAF2*; down: *PLEKHB2*, *RASGRP3*; S6 Table). One gene (*PRAF2*) showed a consistent up-regulation also in the cell line data comprising other cancer entities (CCLE cohort). Enrichment analysis of the 30 genes associated with *TERT* promoter mutations showed overlaps with genes involved in 'cilium organization'; 'plasma membrane bound cell projection assembly' and 'actin cytoskeleton' (S7A Table). To elucidate whether our findings are transferable to brain tumors, we extended our analyses to glioblastoma, in which *TERT* promoter mutations are highly recurrent [3] and diagnostic biomarkers [30]. Analysis of genes associated with *TERT* promoter mutations in 28 bulk tumor samples from the glioblastoma subcohort in PCAWG showed that of the six genes consistently associated with *TERT* promoter mutations in all bulk RNA-Seq cohorts, only *ARL17A* was also upregulated in glioblastoma (S6A Table).

In the context of *TERT* expression, 17 genes were consistently upregulated with *TERT* expression in all seven bulk RNA-seq cohorts, comprising melanoma and a cross-entity cancer cohort, as well as a cross-entity cell line cohort (Table 2, S7 Table). We confirmed that these *TERT* correlates were mostly tumor cell-specific genes within melanoma, as 15 genes were predominantly expressed in malignant tumor cells in our scRNA-seq data (Table 2, S7A and S7H Table). Enrichment analysis of the 17 genes upregulated with *TERT* expression revealed 'DNA repair', 'double-strand break repair', and 'replication fork processing' as relevant processes (Fig 3B, S8A Table). We replicated most of these associations in glioblastoma (Klughammer cohort) where 14 of 17 genes were also positively associated with *TERT* expression, including four genes related to DNA repair (*XRCC2*, *MSH5*, *TRAIP*, *BRIP1*, S6 Table).

To further elucidate tumor cell-specific signaling pathways, we used single-cell RNA-seq data and analyzed the top 100 genes co-expressed with *TERT* mRNA and performed an enrichment analysis with pathways, ontologies and cell type enrichment sets (S8C, S8D Table). Several cellular expression programs were repeatedly enriched, including the extracellular matrix, integrins, stromal cells, and smooth muscle cells (S8B Table).

## Association of *TERT* and mitochondrial variables

To further elucidate the alternative role of TERT beyond telomere elongation in the molecular biology of mitochondria and reactive oxygen species (ROS) [15, 31], we first tested associations of several ROS-related candidate genes with the presence of genetic alterations in *TERT*. However, we did not observe consistent up- or downregulation off the examined genes *NOX1-5*, *DUOX1/2*, *SOD1/2*, *OPA1*, *MFN1/2*, *PINK1*, *FUNDC1*, and *GPX4* in correlation with *TERT* promoter mutations or expression (S5B and S5C Table). Yet, within our discovery approach,

**Table 2. Genes repeatedly associated with *TERT* mRNA in 9 cohorts (S7 Table).**

| Overlap of bulk RNA-seq and CCLE cohort | Overlap with malignant single-cell | Consistent up or downregulated[a] | Function |
|---|---|---|---|
| 17 | 15 | | |
| *FBXW9* | *FBXW9* | Up | Substrate-recognition component of the SCF (SKP1-CUL1-F-box protein)-type E3 ubiquitin ligase complex |
| *KIF14* | *KIF14* | Up | Microtubule motor protein that binds to microtubules with high affinity through each tubulin heterodimer and has an ATPase activity (By similarity), plays a role in many processes like cell division, cytokinesis and also in cell proliferation and apoptosis |
| *KLHL23* | *KLHL23* | Up | Kelch-Like Family Member 23 (no function available) |
| *MSH5* | *MSH5* | Up | Involved in DNA mismatch repair and meiotic recombination processes |
| *PUS7* | *PUS7* | Up | Pseudouridylate synthase that catalyzes pseudouridylation of RNAs, acts as a regulator of protein synthesis in embryonic stem cells by mediating pseudouridylation of RNA fragments derived from tRNAs |
| *SMARCA4* | *SMARCA4* | Up | Involved in transcriptional activation and repression of selected genes by chromatin remodeling (alteration of DNA-nucleosome topology), component of SWI/SNF chromatin remodeling complexes |
| *STIL* | *STIL* | Up | Immediate-early gene, plays an important role in embryonic development as well as in cellular growth and proliferation, its long-term silencing affects cell survival and cell cycle distribution as well as decreases CDK1 activity correlated with reduced phosphorylation of CDK1 |
| *TMEM201* | *TMEM201* | Up | Involved in nuclear movement during fibroblast polarization and migration |
| *TRAIP* | *TRAIP* | Up | E3 ubiquitin ligase required to protect genome stability in response to replication stress |
| *TTK* | *TTK* | Up | Phosphorylates proteins on serine, threonine, and tyrosine, probably associated with cell proliferation |
| *UCKL1* | *UCKL1* | Up | May contribute to UTP accumulation needed for blast transformation and proliferation |
| *XRCC2* | *XRCC2* | Up | Involved in the homologous recombination repair (HRR) pathway of double-stranded |
| *SMG5* | *SMG5* | Up | Plays a role in nonsense-mediated mRNA decay |
| *DNA2* | *DNA2* | Up | Key enzyme involved in DNA replication and DNA repair in nucleus and mitochondriaS1 |
| *ALKBH2* | *ALKBH2* | Up | Dioxygenase that repairs alkylated DNA and RNA containing 1-methyladenine and 3-methylcytosine by oxidative demethylation |
| *BRIP1* | | Down | DNA-dependent ATPase and 5' to 3' DNA helicase required for the maintenance of chromosomal stability |
| *ARGHGEF39* | | Up | Rho Guanine Nucleotide Exchange Factor 3 |

TERT = telomerase reverse transcriptase; CCLE = cancer cell line encyclopedia

Note: Genes are sorted alphabetically and the first 12 genes were also found in glioblastoma

[a]Consistent up- or down-regulation: similar direction of correlation with *TERT* expression in all bulk, single-cell, and cell line cohorts; up = positively correlation, down = negative correlation

we identified a consistent upregulation of *DNA2* with higher *TERT* expression in several bulk tumor cohorts, potentially pointing to a role of *TERT* in DNA repair (Table).

We then analyzed the mutational load specifically in the mitochondria using the cross-entity PCAWG cohort (n = 1200 with data on mitochondrial mutations, S3C and S3E Table). *TERT* promoter mutation status was not associated with nonsynonymous mitochondrial or other mitochondrial mutations (S3C and S3E Table). However, *TERT* expression correlated with fewer, especially nonsynonymous mitochondrial mutations (Wilcoxon p = 0.007, Spearman p = 0.049, coeff = -0.06, S3D Table, Fig 3C) in stark contrast to the increased overall, and mainly nuclear, mutations assessed above (in SKCM metastases, n = 363 with data on overall mutations, Fig 3B). We further tested whether the expression of the mitochondrial targeting

sequence, which is required for the transport of TERT into mitochondria [24], could be affected by *TERT* promoter mutations. Here, we did not detect a change in the proportion of expressed isoforms with mitochondrial targeting sequence (S5A Table) and conclude that *TERT* promoter mutations do not promote or restrain transcripts with mitochondrial targeting (Fig 3C).

## *TERT* alterations in hypoxia and chromothripsis

We next tested whether *TERT* promoter mutations and *TERT* expression correlated with hypoxia and chromothripsis in the PCAWG and SKCM cohorts (S3B–S3E Table). Ragnum hypoxia scores in melanoma patients (SKCM, Wilcoxon p = 0.024, S3C Table) and Buffa scores across entities (PCAWG, Wilcoxon p = 4.678e-21, Fig 3C) were higher in samples carrying a *TERT* promoter mutation. We also observed upregulation of *TERT* expression with all hypoxia scores in the melanoma cohort (S3B Table) and with Buffa hypoxia scores across entities (Wilcoxon p = 7.43e-22). We observed significantly more chromothripsis in samples carrying *TERT* promoter mutations and a higher median *TERT* expression in samples with chromothripsis (PCAWG, S3D, S3E Table).

## Discussion

### *TERT* and ICI therapy

*TERT* promoter mutations have been discussed as potential prognostic factors in metastatic melanoma [5, 6, 8, 26, 27] and as potentially predictive most recently under BRAF/MEK [7] and anti-CTLA4 therapy [9]. TMB is discussed as a biomarker for ICI success [32, 33] and *TERT* promoter mutations showed association with higher TMB [9], however, multivariate approaches controlling for TMB were missing. Our analyses show that, when controlling for mutational load and prior MAPK inhibitor therapy, the association of *TERT* promoter mutations in the tested anti-CTLA4 and non-ICI cohorts was weak or absent. Hence, we believe that the extent to which *TERT* promoter mutations govern survival under anti-CTLA4 therapy is not statistically supported [7, 8]. In addition, since elevated *TERT* expression was not associated with survival, we conclude that melanoma differs from other cancer entities that showed an association between *TERT* levels and prognosis in lung cancer, oral squamous cell carcinoma, and colorectal carcinoma [10–12]. To our best knowledge, TMB, however, was not assessed in these studies, and could possibly be improved by multivariate analyses. A striking correlation of *TERT* expression in our study was a higher estimate of tumor-infiltrating CD4 + T cells, which were previously associated with improved clinical outcome of ICI therapy [34]. There is evidence for a baseline TERT-specific CD4+ T cell immune response, occurring in over 50% of melanoma patients, rising to 80% in ICI responders [35, 36]. Consequently, a higher CD4+ T cell response, possibly explained by an anti-TERT T-helper type 1 cell response, could define a subgroup of patients with better ICI therapy response. However, our analyses of single-cell expression showed that, if malignant cells expressed higher levels of *TERT*, the corresponding CD4+ T cells from these tumors expressed exhaustion markers at higher levels (Fig 1B). Therefore, in *TERT*-high hypoxic melanoma, dysfunctional CD4+ T cells may accumulate employing these immuno-suppressive checkpoints [37]. As previously suspected, this could argue for a defect in anti-TERT CD4+ response during melanoma progression [36] and may explain why in our analysis, patients with high CD4+ T cell estimates and increased *TERT* expression did not show a survival benefit.

## *TERT* expression in metastases from thin primaries

*TERT* promoter mutations were previously described to upregulate *TERT* expression [38, 39] involving the formation of ETS/TCF transcription factor binding sites [2, 38], and have been found in primary and metastatic melanoma. Here, metastatic melanomas showed higher *TERT* expression if they originated from thinner primaries, prompting the question of increased *TERT* expression in efficient early metastasis. This is especially relevant as *TERT* expression has been shown to be involved in most steps of the invasion-metastasis cascade [40]. The local invasion of primary melanoma cells might be regulated through increased *TERT* expression affecting cellular programs, such as NF-kB or metalloproteinases in addition to canonical telomerase activity [41, 42]. As the gene expression enrichment of specifically malignant melanoma cells revealed that *TERT* expression is co-regulated with integrins, TGF-beta, and interleukins, a *TERT* expression shift may be involved in remodeling the extracellular matrix, which is seen as a driving force for cancer stemness [43] and would aid invasion. It remains unclear whether *TERT* expression was elevated already in the thinner primary tumors corresponding to the metastases measured for RNA (Fig 2B). Higher Breslow depth was previously shown to be associated with the presence of *TERT* mutations in primary tumors [5, 26]. However, this association was not apparent in our analysis, probably because only thicker primaries were selected for the RNA-seq experiment in the SCKM cohort.

In conclusion, our data indicate that *TERT* expression may contribute to early metastasis from thin primaries, potentially through extracellular matrix remodeling, without a clear role for promoter mutations in this respect.

## Hotspot mutations, mutational load and DNA repair

Initially, we confirmed the known co-occurrence of *TERT* promoter mutations and hotspot mutations in *BRAF*, *RAS* and *NF1* as previously described for melanoma [5]. While, thus a co-mutation of the *TERT* promoter and members of the MAPK pathway is evident, *TERT* expression varied significantly between mutants, showing lower *TERT* expression in *BRAF* mutants compared to *RAS* mutants. Moreover, our finding that *BRAF* mutants exhibited lower *TERT* expression compared with non-*BRAF* mutants in *TERT* promoter-mutated samples somewhat contradicts the proposed model of *TERT* reactivation in *BRAF*-mutated melanoma samples through acquisition of *TERT* promoter mutations, subsequent binding of TCF, and thus increased *TERT* expression [44]. These contradicting findings are also of great interest for thyroid tumors, as the status of *TERT* promoter mutations together with *BRAF* mutations have been shown to be associated with clinicopathological aggressiveness [45–47]. As our analysis of co-mutations were only available in one cohort, further investigations may improve our understanding of the MAPK pathway and *TERT* activation via *TERT* promoter mutations.

As a source of hotspot mutations, mutational load is also correlated with *TERT* expression in metastatic melanomas here and previously described [9]. The increased co-expression of DNA repair-related genes suggests that *TERT* expression may be involved in establishing genomic stability under replication stress since *TERT* expression regulates the chromatin state and DNA damage responses in normal human fibroblasts [48]. Beyond melanoma, co-expression of *TERT* and DNA repair genes was shared in various cancer entities including GBM, however, common gene expression patterns were not observed in association with *TERT* promoter mutations. To date, studies of *TERT* alterations in GBM have focused on the fairly frequent promoter mutations. Hence, our results may spark interest in assessing *TERT* expression levels in GBM to test sensitivity to DNA damage [49]. This could be relevant in the context of chemo- and radiotherapy independent of *TERT* promoter mutations, since *TERT* expression is also mediated via rearrangements and hypermethylation [50]. Moreover, our results support the view that this non-canonical function of

*TERT* could be especially relevant under hypoxic conditions and after chromothripsis, where *TERT* expression was increased. In line with the proposed concept of a two-step activation of *TERT* [38], the up-regulation of *TERT* expression in later stages of melanoma development could aid the survival of melanoma cells in hypoxic tumors and metastasis to other sites.

### *TERT* and mitochondrial stability

In cancer cells exposed to hypoxia, ROS levels produced by mitochondria are increased [51]. Moreover, *TERT* has previously been shown to be involved in the protection of cells from ROS-mediated oxidative stress, delaying aging processes supposedly via TERT activity in mitochondria [15]. As these effects were described predominantly in non-cancer tissues, we aimed to identify potential mitochondrial correlates of *TERT* expression in cancer and, specifically, in melanoma. While at first, we did not find associations of *TERT* with a set of fifteen ROS-related candidate genes, our less-biased discovery approach identified a consistently upregulated *DNA2*, known to be involved in DNA repair in the nucleus and mitochondria [52]. This is in line with *TERT* translocation into mitochondria to protect mitochondrial DNA from damage and implicates a dedicated repair gene in the described process [15, 24]. The correlation between *TERT* expression and fewer nonsynonymous mitochondrial mutations further supports this rationale. It contrasts though, with the fact that *TERT* expression and *TERT* promoter mutations were associated with more overall mutations in the nuclear genome. Therefore, DNA repair via *TERT* and possibly *DNA2* may be mitochondria-specific. Interestingly, differing mutational processes have already been described in colorectal and gastric cancers, where microsatellite instability was observed in the nuclear, but not in the mitochondrial genome [53]. Thus, it would be highly interesting to explore the cellular localization of TERT and DNA2 in various cancers and in the context of hypoxia.

A potential limitation of our study is that different subtypes of *TERT* promoter mutations may differentially affect *TERT* expression and, therefore clinical factors. However, individual promoter mutations were only available for the SKCM and CCLE cohorts. Indeed, in both cohorts, the C288T mutation is the only mutation showing association with increased *TERT* expression (S5D Table). However, analyzing the different promoter mutations individually, whenever possible, did not reveal any differences in our results (S10 Fig). We strongly believe that, as the assessment of mutation types is currently part of routine testing, the different effects of the subtypes of *TERT* promoter mutations could be taken into account in future studies.

In conclusion, our analyses did not support a significant association between *TERT* genetic alterations and survival in melanoma patients. Hence, its presumed role as a biomarker, especially for anti-CTLA4 or MAPK inhibitor therapy, needs further validation. In addition, our data indicate dysfunctional CD4+ T cells in *TERT* high hypoxic melanoma. Since melanoma metastases with high *TERT* expression were shown to have originated from thinner primary tumors, we conclude that upregulation of *TERT* expression may contribute to metastasis. This is also supported by the involvement of *TERT* expression in the dynamics of the extracellular matrix. We add support to the non-canonical function of *TERT* expression related to mitochondrial and nuclear DNA stability even across cancer entities, especially under hypoxic conditions and after chromothripsis. In summary, our results demonstrate the possible role of *TERT* expression in metastasis formation and invasion in melanoma and highlight its role in DNA repair mechanisms across cancer entities.

## Supporting information

**S1 File. Supplementary material including the supplementary methods, description of supplementary tables and S1-10 Figs.**
(DOCX)

**S1 Table. Multivariate survival analysis.** Multivariate analysis using Cox proportional hazard models for progression free survival (PFS) (S1A Table) and overall survival (OS) (S1B Table) subsetted for ICI therapy type in cohorts Liu, Gide, Hugo, Riaz, Van Allen and SKCM.
(XLSX)

**S2 Table. *TERT* expression and immune signatures.** Significant immune signatures in SKCM associated with *TERT* expression and comparison against other bulk RNA-seq cohorts (Gide, Hugo, Liu, Riaz, Van Allen).
(XLSX)

**S3 Table. TERT with respect to clinical variables.** Wilcoxon tests for group comparisons and Spearman correlations of clinical variables against *TERT* expression and *TERT* promoter mutation in Liu, SKCM and PCAWG cohorts (S3A–S3E Table).
(XLSX)

**S4 Table. TERT in melanoma subtypes.** Mutation subtypes against *TERT* expression (S4A Table) and *TERT* promoter mutation status (S4B Table) in the SKCM melanoma cohort. Effect of hotspot mutations on *TERT* expression in *TERT* promoter mutated samples (S4C Table).
(XLSX)

**S5 Table. *TERT* isoforms and mutation subtypes and influence of *TERT* alterations on ROS.** Influence of TERT promoter mutation on *TERT* isoform on mitochondrial targeting sequence (S5A Table), associations of ROS-related genes with *TERT* expression levels (S5B Table), and *TERT* promoter mutation status (S5C Table) in Gide, Riaz, Hugo, Liu, Van Allen, SKCM, PCAWG, CCLE cohorts, and single-cell. Difference on *TERT* promoter mutation subtypes on *TERT* expression on SKCM and CCLE cohorts (S5D Table).
(XLSX)

**S6 Table. Genetic correlates with *TERT* promoter mutations.** Genes associated with *TERT* promoter mutation status for the Van Allen, SKCM, PCAWG, and CCLE cohorts, and overlap of cohorts (S6A–S6E Table).
(XLSX)

**S7 Table. Genetic correlates with *TERT* expression.** Genes associated with *TERT* expression in each cohort (Gide, Riaz, Van Allen, Hugo, Liu, SKCM, PCAWG, CCLE, single-cell, and Klughammer) and overlap of cohorts (S7A–S7K Table).
(XLSX)

**S8 Table. Enrichment analysis.** Enrichment analysis of 17 genes associated with *TERT* expression and 30 genes associated with *TERT* promoter mutation in bulk tumors and cell lines (S8A Table), and for the top 100 genes associated with *TERT* expression in single-cells (S8B Table).
(XLSX)

## Acknowledgments

We thank Udo Stenzel for help with the data analysis.

## Author Contributions

**Conceptualization:** Christina Katharina Kuhn, Susanne Horn.

**Data curation:** Christina Katharina Kuhn, Sven-Holger Puppel, Susanne Horn.

**Formal analysis:** Christina Katharina Kuhn, Susanne Horn.

**Funding acquisition:** Christina Katharina Kuhn, Torsten Schöneberg, Dirk Schadendorf, Susanne Horn.

**Investigation:** Christina Katharina Kuhn, Susanne Horn.

**Methodology:** Christina Katharina Kuhn, Susanne Horn.

**Project administration:** Torsten Schöneberg, Dirk Schadendorf, Susanne Horn.

**Resources:** Sven-Holger Puppel, Susanne Horn.

**Software:** Susanne Horn.

**Supervision:** Torsten Schöneberg, Susanne Horn.

**Validation:** Christina Katharina Kuhn, Susanne Horn.

**Visualization:** Christina Katharina Kuhn, Susanne Horn.

**Writing – original draft:** Christina Katharina Kuhn, Jaroslawna Meister, Sophia Kreft, Mathias Stiller, Anne Zaremba, Björn Scheffler, Vivien Ullrich, Susanne Horn.

**Writing – review & editing:** Christina Katharina Kuhn, Jaroslawna Meister, Sophia Kreft, Mathias Stiller, Sven-Holger Puppel, Anne Zaremba, Björn Scheffler, Vivien Ullrich, Susanne Horn.

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
