## [Decision Letter · Decision Letter 0]

6 Mar 2023

PONE-D-23-01824*TERT* expression is associated with metastasis from thin primaries, exhausted CD4+ T cells in melanoma and with DNA repair across cancer entitiesPLOS ONE

Dear Dr. Kuhn,

Thank you for submitting your manuscript to PLOS ONE. After careful consideration, we feel that it has merit but does not fully meet PLOS ONE’s publication criteria as it currently stands. Therefore, we invite you to submit a revised version of the manuscript that addresses the points raised during the review process.

We look forward to receiving your revised manuscript.

Kind regards,

Avaniyapuram Kannan Murugan, M.Phil., Ph.D.

Academic Editor

PLOS ONE

Journal Requirements:

"The authors declare that they have no conflict of interest."

3. We note that Figures 1, 2, 3 and S1 in your submission contain copyrighted images. All PLOS content is published under the Creative Commons Attribution License (CC BY 4.0), which means that the manuscript, images, and Supporting Information files will be freely available online, and any third party is permitted to access, download, copy, distribute, and use these materials in any way, even commercially, with proper attribution. For more information, see our copyright guidelines: http://journals.plos.org/plosone/s/licenses-and-copyright.

a. You may seek permission from the original copyright holder of Figures 1, 2, 3 and S1 to publish the content specifically under the CC BY 4.0 license. 

Additional Editor Comments;

1. Raised critiques to be addressed in a point-by-point manner before consideration, this would also  substantially improve the manuscript.

2. Care is necessary in addressing particularly the reviewer #2 and the same is critical for publication.

3. Consider discussing if mutation in TERT could have any similar impact as expression and discuss the same with melanoma and others including thyroid by citing these articles PMID: 23766237; PMID: 24617711; PMID: 25024077.

Reviewers' comments:

Reviewer's Responses to Questions

**Comments to the Author**

1. Is the manuscript technically sound, and do the data support the conclusions?

Reviewer #1: Partly

Reviewer #2: Partly

Reviewer #3: Yes

2. Has the statistical analysis been performed appropriately and rigorously? 

Reviewer #1: Yes

Reviewer #2: Yes

Reviewer #3: Yes

3. Have the authors made all data underlying the findings in their manuscript fully available?

Reviewer #1: Yes

Reviewer #2: Yes

Reviewer #3: Yes

4. Is the manuscript presented in an intelligible fashion and written in standard English?

Reviewer #1: Yes

Reviewer #2: Yes

Reviewer #3: Yes

5. Review Comments to the Author

Reviewer #1: It would help to appreciate the importance of the identified TERT mRNA gene associations and the directions (up/down regulated) better if the authors could add the corresponding gene expression levels (raw expression counts normalized to logCPM (counts per million)) and the logFC (fold changes).

Reviewer #2: In this manuscript, Kuhn et al. claimed that TERT alterations were not consistently

associated with survival in melanoma cohorts under immune checkpoint inhibition. The presence of CD4+ T cells increased with TERT expression and correlated with the expression of exhaustion markers. While the frequency of promoter mutations did not change with Breslow thickness, TERT expression was increased in metastases arising from thinner primaries. Increased TERT expression was related to mitochondrial DNA stability and nuclear DNA repair during chromothripsis (PCAWG cohort) and under hypoxic conditions (PCAWG and SKCM cohorts). This should be a clear story, but authors tried to fit all the data into one story and made it so hard to understand and audiences were keeping lost in the irrelevant description.

Major points:

1. TERT promoter mutations were not always be associated with TERT expression in all cohorts. This made things much more complicated and confusing. If some mutations led to high TERT expression, while other mutations led to low TERT expression or irrelevant, authors should take this into account and some non-significant differences may actually significant.

2. It is so hard to understand the results parts and figures. They seem completely separated. I can’t even find where the figures were mentioned in the result parts (fig. 1B was even only in the discussion part). My suggestion is following the figures, which are much clearer than the results description.

Minor points:

1. There should be an abbreviation chart or explain every abbreviation when they appear first time.

2. Page 6, single-cell RNA-seq data were mentioned. However, authors failed to describe the results.

3. Page 7, in first line, “was not associated with nonsynonymous”, is that a typo or the result in page 6 was wrong?

4. Discussion should mainly focus on authors’ data.

Reviewer #3: The manuscript-“TERT expression is associated with metastasis from thin primaries, exhausted CD4+ T cells in melanoma, and with DNA repair across cancer entities” by Kuhn et al, is an interesting study adding new insights into melanoma metastasis that is attributed to TERT expression. The manuscript is well-written, and the methodology is adequately described. The overall results presented in the manuscript are logically described. The choice of melanoma cohorts, cell lines as well as single-cell RNA seq data analysis look appropriate in context to TERT mutations. Overall, the results are clear and compelling with a well-structured possible role of TERT in mitochondrial and nuclear DNA stability.

I would recommend this manuscript for publication.

6. PLOS authors have the option to publish the peer review history of their article (what does this mean?). If published, this will include your full peer review and any attached files.

Reviewer #1: No

Reviewer #2: No

Reviewer #3: No

---

## [Author Response · Author response to Decision Letter 0]

25 Apr 2023

We would like to thank the editorial team and the reviewers for their valuable comments and in-depth review of our manuscript. We are convinced that the critiques have had a great impact on the strengthening of the quality of our work. We have carefully addressed all the criticisms, especially those of reviewer #2, in the attached file "Response to Reviewers".

---

## [Decision Letter · Decision Letter 1]

20 Jun 2023

*TERT* expression is associated with metastasis from thin primaries, exhausted CD4+ T cells in melanoma and with DNA repair across cancer entities

PONE-D-23-01824R1

Dear Dr. Katharina Kuhn,

We’re pleased to inform you that your manuscript has been judged scientifically suitable for publication and will be formally accepted for publication once it meets all outstanding technical requirements.

Kind regards,

Avaniyapuram Kannan Murugan, M.Phil., Ph.D.

Academic Editor

PLOS ONE

Additional Editor Comments (optional):

Reviewers' comments:

Reviewer's Responses to Questions

**Comments to the Author**

1. If the authors have adequately addressed your comments raised in a previous round of review and you feel that this manuscript is now acceptable for publication, you may indicate that here to bypass the “Comments to the Author” section, enter your conflict of interest statement in the “Confidential to Editor” section, and submit your "Accept" recommendation.

Reviewer #1: All comments have been addressed

2. Is the manuscript technically sound, and do the data support the conclusions?

Reviewer #1: Yes

3. Has the statistical analysis been performed appropriately and rigorously? 

Reviewer #1: Yes

4. Have the authors made all data underlying the findings in their manuscript fully available?

Reviewer #1: Yes

5. Is the manuscript presented in an intelligible fashion and written in standard English?

Reviewer #1: Yes

6. Review Comments to the Author

Reviewer #1: (No Response)

7. PLOS authors have the option to publish the peer review history of their article (what does this mean?). If published, this will include your full peer review and any attached files.

Reviewer #1: No

---

## [Editor Report · Acceptance letter]

27 Jun 2023

PONE-D-23-01824R1 

*TERT* expression is associated with metastasis from thin primaries, exhausted CD4+ T cells in melanoma and with DNA repair across cancer entities 

Dear Dr. Kuhn:

I'm pleased to inform you that your manuscript has been deemed suitable for publication in PLOS ONE. Congratulations! Your manuscript is now with our production department. 

Kind regards, 

on behalf of

Dr. Avaniyapuram Kannan Murugan 

Academic Editor

PLOS ONE